 # CodeLutra: Boosting LLM Code Generation via Preference-Guided Refinement

**Leitian Tao**                                                                      *leitiantao@cs.wisc.edu*
*University of Wisconsin-Madison*

**Xiang Chen**                                                                       *xiangche@adobe.com*
*Adobe Research*

**Tong Yu**                                                                          *tyu@adobe.com*
*Adobe Research*

**Tung Mai**                                                                         *tumai@adobe.com*
*Adobe Research*

**Ryan A. Rossi**                                                                    *ryrossi@adobe.com*
*Adobe Research*

**Yixuan Li**                                                                        *sharonli@cs.wisc.edu*
*University of Wisconsin-Madison*

**Saayan Mitra**                                                                     *smitra@adobe.com*
*Adobe Research*

**Reviewed on OpenReview:** *https://openreview.net/forum?id=IGsEgWM4to*

## Abstract

Large Language Models (LLMs) have revolutionized code generation but require significant resources and tend to over-generalize, limiting their task-specific efficiency. Fine-tuning smaller, open-source LLMs is a cost-effective alternative, yet standard supervised approaches rely solely on correct examples, overlooking valuable insights from failures. We explore recent developments in preference-based post-training to code generation, leveraging both correct and incorrect code attempts. Instead of solely relying on correct examples, our framework CODELUTRA iteratively refines the model by comparing successful and unsuccessful outputs, thereby more accurately aligning the generated code with desired outcomes. This process narrows the performance gap with state-of-the-art, larger models, without requiring massive datasets or auxiliary models. For example, on a challenging data science coding task, using only 500 samples improved Llama-3-8B's accuracy from 28.2% to 48.6%, approaching GPT-4's level. CODELUTRA provides a scalable and efficient strategy for high-quality code generation, helping smaller open-source models narrow the performance gap with state-of-the-art closed-source systems.

## 1 Introduction

Large language models (LLMs) have revolutionized numerous domains, consistently delivering great performance across different tasks (Brown et al., 2020; Achiam et al., 2023; Anthropic, 2023; OpenAI, 2023). Among these applications, code generation stands out as particularly promising. Models pre-trained on ex-

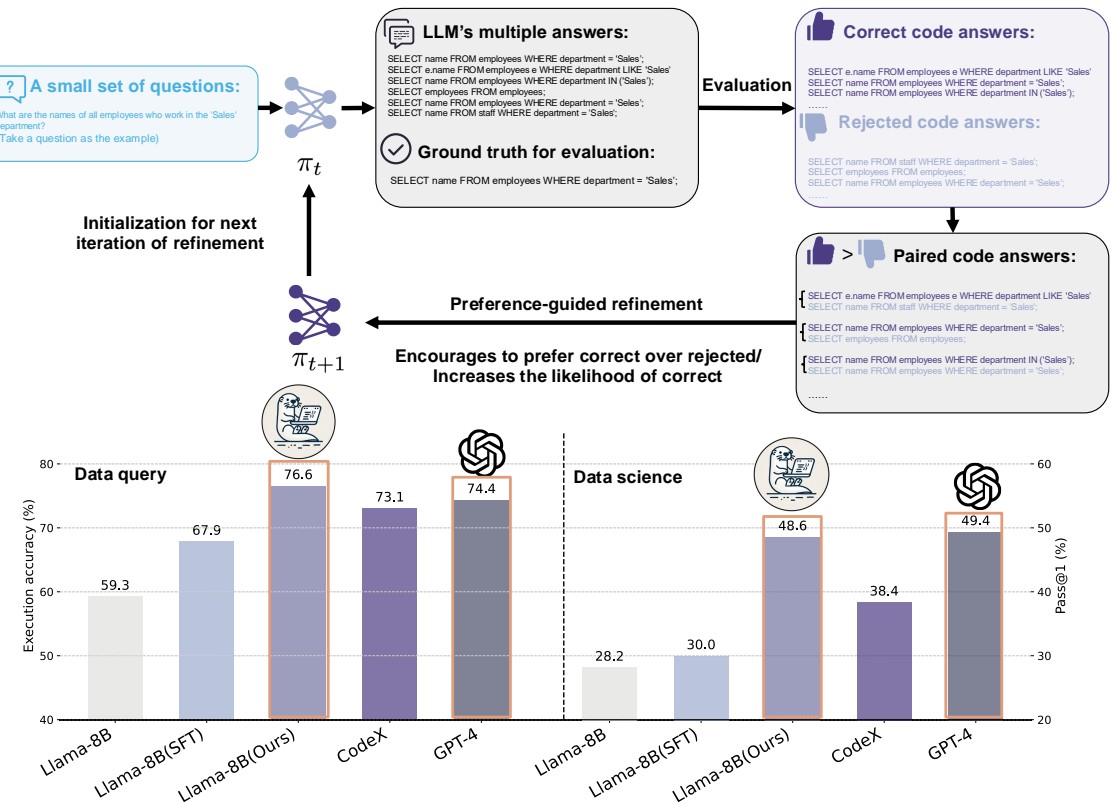

Figure 1: The proposed CODELUTRA framework (see Section 4) and performance comparison on different code generation tasks. The orange dashed box highlights our framework's performance relative to GPT-4.

tensive code repositories have demonstrated an impressive capability to solve diverse programming challenges (Li et al., 2022; Zheng et al., 2023; Chen et al., 2021b; Wang et al., 2023).

Deploying ultra-large closed-source models like GPT-4 (OpenAI, 2023) for code generation is challenging due to their enormous resource demands and limited customization options. Fine-tuning smaller, open-source LLMs offers a more flexible and cost-effective alternative, yet code generation requires more than syntactical accuracy—it demands logical depth and domain-specific understanding. While basic supervised fine-tuning (SFT) on correct code examples can help, it yields only incremental gains and often falls short of state-of-the-art performance. This is because SFT alone cannot leverage the model's own mistakes, limiting its ability to adapt and improve in diverse programming environments. We illustrate this in Figure 1, where SFT provides a minor boost in performance on the challenging data science task, with Llama-3-8B's Pass@1 improving from 28.2% to 30.0%. A notable gap remains between the fine-tuned model and GPT-4, which leads with a Pass@1 score of 49.4%. To address the issue, prior solutions collect additional training data by leveraging more powerful LLMs (Shen et al., 2023; Luo et al., 2023) or by collecting external datasets from public repositories (Lozhkov et al., 2024; Muennighoff et al., 2023). *But the big elephant in the room still remains—how much can we bridge the gap by maximizing the utility of existing data and model at hand?*

To address this challenge, we introduce CODELUTRA, a framework that iteratively improves the performance of a given LLM *without relying on vast external datasets or larger auxiliary models*. Unlike traditional fine-tuning methods that rely exclusively on correct code solutions, our framework CODELUTRA leverages both successful and failed code generated by the model itself, creating self-generated comparative data. While prior works explored using execution results to enhance code generation, they often require extensive datasets to train reward models for feedback (Le et al., 2022; Shojaee et al., 2023) or rely on prompt engineering with ultra-large LLMs (Chen et al., 2023; Ni et al., 2024). In contrast, our work collects successful and failed attempts and enables models to iteratively learn, distinguish, and improve from these examples. Our approach demonstrates that even with limited data at hand, substantial gains in code generation quality can be achieved, closing the gap between smaller fine-tuned models and the top-tier LLMs. To harness

the comparative data, a key component of CODELUTRA is the preference-guided refinement mechanism that compares correct and incorrect code snippets, iteratively refining the model's understanding of code quality. With each iteration, the model generates code solutions, evaluates their correctness, and updates its parameters based on the evolving preference dataset. This process allows for continuous improvements of the base model, making the framework effective even with limited initial data (e.g., a few hundred samples). The failed code attempts are invaluable for refining models, since they provide concrete examples of common errors and enable the model to learn strategies for avoiding similar mistakes in future generations.

We comprehensively evaluate the effectiveness of CODELUTRA on challenging data query and data science tasks, where the LLM is tasked with generating the correct SQL or Python code to solve a given problem. We compare CODELUTRA with 13 open-source and closed-source LLMs that are competitive in code generation. Notably, on the data query task, *our framework allows Llama-3-8B (Dubey et al., 2024) to achieve an execution accuracy of 76.6%, which exceeds GPT-4's 74.4%.* Under a challenging data science task, we find that using just 500 samples improved Llama-3-8B from an accuracy of 28.2% to 48.6%, approaching the performance of GPT-4. This demonstrates that CODELUTRA achieves strong results even with a limited number of high-quality annotations. Moreover, we observe the consistent performance gains of CODELUTRA on different base models, including Gemma-7B (Team et al., 2024) and StarCoder-7B (Dai & Kumar, 2023). These findings highlight the potential of CODELUTRA in closing the performance gap between open-source and closed-source models. To summarize our key contributions:

1. We explore preference-based learning for code generation through CODELUTRA, which leverages self-generated comparative data from both successful and failed code attempts, enabling low-performing models to rival top-tier solutions without external datasets or ultra-large LLMs' feedback.

2. We conduct comprehensive evaluations, comparing CODELUTRA against 13 competitive LLMs specializing in code generation. Results demonstrate CODELUTRA outperforms both standard fine-tuned LLMs and existing cutting-edge closed-source LLMs.

3. We conduct the in-depth analyses to understand the contribution of failed attempts and likelihood regularization for CODELUTRA, and reveal CODELUTRA improves the performance via reducing syntax errors and improving incorrect answers across iterations.

## 2 Related Work

**Preference learning for LLMs.** Preference learning guides language models toward outputs aligned with human preferences. Approaches like RLHF (Ziegler et al., 2019; Ouyang et al., 2022) effectively align LLMs but face inefficiencies and hyperparameter sensitivity (Christiano et al., 2017; Bai et al., 2022). Recent works focus on closed-form loss functions using offline preference data, such as DPO (Rafailov et al., 2023) and its extensions (Liu et al., 2023b; Ethayarajh et al., 2024). Iterative DPO, where models generate their own preference data, shows strong performance in the alignment and math reasoning tasks (Liu et al., 2024; Xiong et al., 2024; Yuan et al., 2024; Pang et al., 2024). Our work represents the first comprehensive study on iterative preference optimization specifically tailored for code generation, achieving GPT-4-level results with significantly fewer resources. While previous approaches (Le et al., 2022; Shojaee et al., 2023; Ni et al., 2024; Chen et al., 2024) leverage execution results predominantly to refine incorrect code until a correct solution emerges, they often rely heavily on complex reward signals from critic models (Le et al., 2022; Shojaee et al., 2023) or sophisticated prompt engineering (Chen et al., 2024; Ni et al., 2024). The key novelty of our method lies in using execution outcomes to probabilistically differentiate correct from incorrect generations via a Bradley–Terry model, representing a fundamental conceptual shift from existing approaches. This conceptual shift allows our framework to remain simpler, more generalizable, and less dependent on proprietary large-scale models: rather than crafting intricate instructions tailored to advanced architectures or relying on external ground truths like unit tests, we directly let the model learn from failures by contrasting them with successes, thereby improving its coding proficiency over iterative training cycles. Compared to other methods, which introduce complexities that may not scale down to smaller models or open-source environments, our minimalistic strategy consistently demonstrates strong performance.

**LLMs for code generation.** LLMs trained on large code corpora excel in tasks like code generation (Chen et al., 2021c; Zhang et al., 2022), program repair (Xia & Zhang, 2022), and software testing (Chen et al., 2023). Models like WIZARDCODER (Luo et al., 2023) and DS-CODER (Li et al., 2023b) enhance these capabilities through repository-level organization and retrieval-augmented techniques (Lu et al., 2022). Instruction fine-tuning, as in CODEINSTRUCT (Li et al., 2023a), further improves alignment with human coding preferences. Unlike existing methods that rely on program execution results (Le et al., 2022; Shojaee et al., 2023; Ni et al., 2024; Zhang et al., 2024), which depend on complex reward models or elaborate prompt engineering, our novel approach iteratively refines small LLMs using self-generated preference data, completely eliminating the need for external datasets and larger models. Additionally, we employ a unification of SFT and preference learning through our dual-loss approach, maximizing the likelihood of correct solutions in one stage. Furthermore, although theoretical insights into dual loss objectives have been discussed in other work (Liu et al., 2024), our approach distinctively refines and applies these principles to the uniquely challenging domain of code generation—showing that the integration of execution feedback into a direct, probabilistic learning framework can yield practical gains without the pitfalls of overly elaborate interventions. While prior iterative reasoning methods excel at mathematical tasks (Pang et al., 2024), code generation poses distinct challenges due to strict syntactic and semantic correctness requirements. In mathematical reasoning, a correct final output does not necessarily guarantee an entirely accurate answer (Wang et al., 2024). However, in code generation, even minor errors in the code can result in completely incorrect execution outcomes. The novelty of our proposed framework, CODELUTRA, lies in its unique integration of both successful and failed attempts using a Bradley–Terry model, enabling the refinement of outputs without any reliance on explicit feedback or ground-truth labels. By generalizing from errors and iteratively improving attempts, CODE-LUTRA consistently reduces syntactic and logical flaws while maintaining the quality of correct solutions. Empirically, it matches or even surpasses advanced models like GPT-4, advancing the democratization of high-quality code generation in a domain with strict correctness requirements.

## 3 Preliminaries

**LLMs for code generation.** LLMs are pre-trained on diverse datasets encompassing both natural and programming languages. In code generation tasks, an LLM receives a prompt—such as a natural language description, and generates the corresponding code by predicting the next token in the sequence. Formally, code generation is modeled as the conditional probability of a code sequence $y = (y_1, y_2, \ldots, y_T)$ given an input prompt $x$:

$$P(y|x) = \prod_{t=1}^{T} P(y_t|y_{<t}, x), \tag{1}$$

where $x$ is the input prompt, $y$ is the generated code sequence of length $T$, and $y_{<t} = (y_1, y_2, \ldots, y_{t-1})$ represents the tokens generated before time step $t$.

**Supervised fine-tuning of LLMs on task-specific dataset.** Pre-trained LLMs can be suboptimal on task-specific dataset, necessitating fine-tuning. We consider a task-specific dataset $\mathcal{D} = \{(x_i, y_i)\}_{i=1}^{n}$ containing $n$ examples, where each pair $(x_i, y_i)$ represents an input prompt $x_i$ and its corresponding target code $y_i$. Supervised fine-tuning (SFT) adjusts the model's parameters $\theta$ by maximizing the likelihood of generating correct code sequences $y_i$. The loss is defined as:

$$\pi_{\text{SFT}} = \operatorname{argmax}_{\pi_\theta} \mathbb{E}_{(x_i, y_i) \sim \mathcal{D}} \left(\log \pi_\theta\left(y_i | x_i\right)\right). \tag{2}$$

**Verification of code correctness.** We verify the correctness of the generated code by running it with the same inputs as the reference code and ensuring that both produce identical outputs.

**Limitations of SFT for code generation.** Code generation demands not just syntactical correctness but also a deep understanding of logical and domain-specific nuances, which complicates model training. *A major limitation of SFT is that it solely maximizes the likelihood of providing correct code, which restricts the model's ability to learn from its own mistakes.* Since the training process focuses exclusively on provided examples, the LLM doesn't receive the gradient from incorrect or suboptimal code. For instance, if the model

only predicts wrongly in the final token in a code snippet, the overall probability $P(y|x)$ might still remain high as the preceding tokens are correct. Correspondingly, the SFT loss is very small in that case. However, unlike natural language generation where minor errors might still preserve meaning, in code generation, this single erroneous token can render the entire code nonfunctional or introduce subtle bugs, significantly affecting execution correctness despite a high likelihood score. This binary nature of code correctness—where code either executes perfectly or fails completely—creates a unique challenge that standard SFT approaches struggle to address. This reliance on only correct examples limits the model's capacity to identify and recover from errors, reducing its ability to handle more complex or nuanced coding tasks. This motivates our framework CODELUTRA, which leverages both successful and failed code generation attempts.

## 4    CodeLutra

In this section, we introduce CODELUTRA, a framework designed to comparatively learn from both correct and incorrect code generations. CODELUTRA delivers substantial performance gains, achieving performance comparable to more advanced models like GPT-4 (OpenAI, 2023), even with limited initial data. We summarize our algorithm in implementation in the Algorithm 1.

**Initialization.**    We start with an initial base model, denoted as $\pi_0$, which serves as the starting point for our iterative refinement process. We are provided with an initial training set $\mathcal{D} = \{(x_i, y_i)\}_{i=1}^n$, where each $x_i$ is a natural language query, and $y_i$ is the corresponding ground truth code solution. Starting with a modestly performing model allows us to clearly observe improvements attributable to the CODELUTRA framework, ensuring that enhancements result from our methodology rather than inherent model capabilities. Note that at initialization, we only have the correct codes in hand. We describe how to obtain incorrect codes to serve the model refinement.

**Generating correct and failed code.**    At each iteration $t$, the model $\pi_t$ generates $M$ code responses for each input $x_i \in \mathcal{D}$: $\hat{y}_i^m \sim \pi_t(x_i)$ for $m \in \{1, 2, \ldots, M\}$. These responses are executed to determine correctness, with successful outputs categorized into $Y_i^{(c)}$ and failures into $Y_i^{(r)}$. This process introduces output diversity and enables the construction of the preference dataset.

**Preference dataset construction.**    A core aspect of our framework is constructing a preference dataset $\mathcal{D}_t$ at each iteration $t$, capturing the relative quality of generated code. For each input $x_i$, $K$ preference pairs are created by pairing one correct code $\hat{y}_i^{c_k} \in Y_i^{(c)}$ with one rejected code $\hat{y}_i^{r_k} \in Y_i^{(r)}$. Sampling with replacement is used if either set contains fewer than $K$ responses. The resulting dataset, $\mathcal{D}_t = \{(x_i, \hat{y}_i^{c_k}, \hat{y}_i^{r_k})\}$, contains $n \times K$ triplets, where $n$ is the size of the initial dataset. This evaluation mechanism offers clear feedback, particularly syntax and execution errors common in code generation tasks, and enables the construction of preference dataset.

**Preference-guided refinement.**    While one can directly employ a preference optimization approach like DPO (Rafailov et al., 2023) on our curated dataset $\mathcal{D}_t$, this approach presents a notable limitation due to its tendency to decrease the likelihood of both correct and rejected code during training. This is evidenced by the dashed lines in Figure 2, which is also observed in Pal et al. (2024); Feng et al. (2024); Liu et al. (2024). This occurs because DPO relies on proxy rewards from limited data, leading to misdirected gradient updates that cause the model to conservatively reduce likelihood for both response types (Liu et al., 2024). This diminishing likelihood can significantly impact our framework, especially since our correct code are critical for successfully

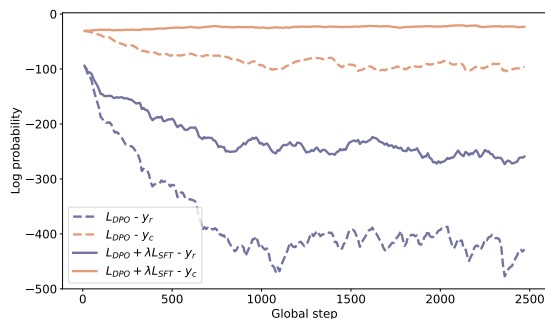

Figure 2: Effect of SFT loss to keep the likelihood of correct answers stable.

solving the assigned tasks. It is crucial, therefore, to prioritize the likelihood of correct solutions to the task at hand.

To address this limitation, we employ a dual-loss function integrating DPO with an additional term, which regularizes the training to prevent decreasing likelihood (Liu et al., 2024). Specifically, we maximize the likelihood of correct solutions in dataset $\mathcal{D}_t^c = \{(x_i, \hat{y}_i^{c_k}) \mid \text{ for all } x_i \in \mathcal{D} \text{ and } k \in [K]\}$. The overall loss function is defined as:

$$\pi_{t+1} = \operatorname{argmin}_{\pi_\theta} \left[ \underbrace{-\mathbb{E}_{(x_i, y_i^c, y_i^r) \sim \mathcal{D}_t} \left[ \log \sigma \left( \beta \left( \log \frac{\pi_\theta(y_i^c|x_i)}{\pi_t(y_i^c|x_i)} - \log \frac{\pi_\theta(y_i^r|x_i)}{\pi_t(y_i^r|x_i)} \right) \right) \right]}_{\text{compare correct and incorrect solutions}} \right.$$
$$\left. - \underbrace{\lambda \mathbb{E}_{(x_i, y_i^c) \sim \mathcal{D}_t} \left( \log \pi_\theta \left( y_i^c|x_i \right) \right)}_{\text{maximize likelihood for correct codes}} \right], \tag{3}$$

where $\lambda$ is a hyperparameter balancing the contributions of the DPO and SFT losses. The first term facilitates preference-based fine-tuning by optimizing the model to favor correct over incorrect code. Concurrently, the second term enhances the likelihood of generating correct solutions directly to avoid the log probability decreasing (see solid line in Figure 2). This dual-loss approach ensures that the model not only ranks correct solutions higher but also increases their generation probability, leading to more consistent high-quality code outputs. We verify the effectiveness of the dual loss empirically in Section 5.3. The refinement process continues until the improvement between consecutive iterations $\pi_t$ and $\pi_{t+1}$ becomes marginal, indicating convergence.

To our knowledge, this is the first work to establish a direct link between iterative preference learning and code generation, where execution results serve as the natural source of feedback. Unlike prior studies that rely on model-generated rewards or other large models' feedback (Chen et al., 2024; Xiong et al., 2024; Yuan et al., 2024; Pang et al., 2024; Xie et al., 2024), we leverage execution correctness as a key preference indicator, uniquely suited to code tasks.

## 5 Experiments

### 5.1 Experimental Setup

**Tasks.** To evaluate the effectiveness of our proposed framework, we perform experiments on two tasks: **Data Query** and **Data Science**. Both tasks reflect common and practical challenges in fields such as business, healthcare, and scientific computing, where precise code generation is critical for solving data-related problems while receiving limited attention. We provide more details and examples of the two tasks in the Appendix A.3.

**Datasets.** We conduct our experiments on two cross-domain datasets for data query, *Spider* (Yu et al., 2018) and *BIRD* (Li et al., 2024), and a data science dataset, *DS-1000* (Lai et al., 2023). We provide more details in the Appendix A.2.

**Metrics.** For the *Data Query* task, we adopt the metrics introduced by Yu et al. (2018): **Execution Accuracy (EX)**, which measures whether the SQL query execution result matches the expected output, and **Exact Match (EM)**, which evaluates whether the generated SQL query exactly matches the reference query in both structure and semantics. For the *Data Science* task, we use **pass@1**, following Lai et al. (2023), which indicates the percentage of correct solutions generated by the model on the first attempt.

**Baselines.** To evaluate the effectiveness of our method, we compare it against three categories of baselines. For a fair comparison, all reported results are based on the same prompt. We report the details of baseline setup in Appendix A.6.

- *Open-source LLMs:* We benchmark our method against competitive open-source LLMs, including models pre-trained on general datasets such as **Llama-3-8B** (Dubey et al., 2024), **Gemma-7B** (Team et al., 2024), and **Llama-3-70B-Instruct** (Dubey et al., 2024). Additionally, we compare against LLMs pre-

---

**Algorithm 1** CODELUTRA

---

**Require:** Training set $\mathcal{D} = \{(x_i, y_i)\}_{i=1}^n$; initial base model $\pi_0$; responses per input $M$; preference pairs $K$; iterations $T$; hyper-parameter $\lambda$

**Ensure:** Fine-tuned model $\pi_T$

1: **for** $t \leftarrow 0$ **to** $T - 1$ **do**
2:     Initialize preference dataset $\mathcal{D}_t \leftarrow \emptyset$
3:     **for all** $x_i \in \mathcal{D}$ **do**
4:         Initialize chosen code set $Y_i^{(c)} \leftarrow \emptyset$
5:         Initialize rejected code set $Y_i^{(r)} \leftarrow \emptyset$
6:         **for** $k \leftarrow 1$ **to** $M$ **do**
7:             Generate response $\hat{y}_i^k \sim \pi_t(x_i)$
8:             **if** execution result of $\hat{y}_i^k$ matches $y_i$ **then**
9:                 Add $\hat{y}_i^k$ to $Y_i^{(c)}$
10:            **else**
11:                Add $\hat{y}_i^k$ to $Y_i^{(r)}$
12:            **end if**
13:         **end for**
14:         **for** $k \leftarrow 1$ **to** $K$ **do**
15:             Randomly sample $\hat{y}_i^{c_k}$ from $Y_i^{(c)}$ (with replacement if $|Y_i^{(c)}| < K$)
16:             Randomly sample $\hat{y}_i^{r_k}$ from $Y_i^{(r)}$ (with replacement if $|Y_i^{(r)}| < K$)
17:             Add $(x_i, \hat{y}_i^{c_k}, \hat{y}_i^{r_k})$ to $\mathcal{D}_t$
18:         **end for**
19:     **end for**
20:     Update model $\pi_{t+1}$ by minimizing the combined loss:

$$\pi_{t+1} = \text{argmin}_{\pi_\theta} \left[ \underbrace{-\mathbb{E}_{(x_i, y_i^c, y_i^r) \sim \mathcal{D}_t} \left[ \log \sigma \left( \beta \left( \log \frac{\pi_\theta(y_i^c | x_i)}{\pi_t(y_i^c | x_i)} - \log \frac{\pi_\theta(y_i^r | x_i)}{\pi_t(y_i^r | x_i)} \right) \right) \right]}_{\text{compare correct and incorrect solutions}} \right.$$

$$\left. - \underbrace{\lambda \mathbb{E}_{(x_i, y_i^c) \sim \mathcal{D}_t} \left( \log \pi_\theta \left( y_i^c | x_i \right) \right)}_{\text{maximize likelihood for correct codes}} \right], \tag{4}$$

21: **end for**

---

trained specifically on coding datasets, such as **Codellama-7B** (Roziere et al., 2023), **StarCoder-7B** (Lozhkov et al., 2024), and **Codestral-22B** (Mistral AI, 2024).

- *Fine-tuned LLMs:* As supervised fine-tuning on domain-specific datasets is a popular and effective way to improve LLMs' corresponding performance, we also report the performance of fine-tuned LLMs using standard supervised fine-tuning on the ground-truth solutions in training data (Raffel et al., 2020).

- *Closed-source LLMs:* We provide the performance of advanced closed-source LLMs, including **Codex** (Chen et al., 2021a), **ChatGPT** (Ouyang et al., 2022), and **GPT-4** (OpenAI, 2023).

**Experimental setup.** For main results, we apply our framework to the **Llama-3-8B** base model (Dubey et al., 2024), denoted as $\pi_0$ (see Section 5.2 for more backbone results). We use a zero-shot prompt containing the question along with reference information. For different answer collections, we employ the best-of-$n$ strategy by sampling 10 responses at the temperature of 1.0. We train 1 epoch per iteration and perform four iterations in total, resulting in models $\{\pi_1, \pi_2, \pi_3, \pi_4\}$. For verification of code correctness, we employ task-specific approaches: For data query tasks, we execute the generated SQL queries against the same database schema as the reference solutions and directly compare execution results, ensuring functional equivalence regardless of syntactic differences. For data science tasks, we leverage the DS-1000 evaluation framework with standardized test cases that verify functional equivalence between generated and reference code. These

Table 1: The Execution Accuracy (EX), Exact Match (EM), and Pass@1 for different kinds of models on SPIDER, BIRD, and DS1000. We show the base model ($\pi_0$) without fine-tuning and the model trained with CODELUTRA in different iteration ($\{\pi_1, \pi_2, \pi_3, \pi_4\}$). For fair comparison, all reported results in the table use the same prompt. **Boldface** highlight GPT-4 and our results.

| Models | Spider | | BIRD | | DS1000 |
|---|---|---|---|---|---|
| | EX | EM | EX | EM | Pass@1 |
| *Open-source LLMs* | | | | | |
| Llama-3-8B (Dubey et al., 2024) | 59.3 | 55.1 | 22.3 | 19.5 | 28.2 |
| Codellama-7B (Roziere et al., 2023) | 57.0 | 51.4 | 24.4 | 18.7 | 25.6 |
| StarCoder-7B (Lozhkov et al., 2024) | 61.2 | 58.6 | 25.7 | 23.0 | 26.8 |
| Gemma-7B (Team et al., 2024) | 49.9 | 46.7 | 21.2 | 19.1 | 24.2 |
| Codestral-22B (Mistral AI, 2024) | 71.3 | 69.6 | 42.5 | 39.9 | 35.8 |
| Llama-3-70B-Instruct (Dubey et al., 2024) | 68.7 | 65.4 | 41.2 | 39.3 | 36.4 |
| *Fine-tuned LLMs* | | | | | |
| Llama-3-8B (Dubey et al., 2024) | 67.9 | 64.7 | 35.6 | 30.7 | 30.0 |
| Codellama-7B (Roziere et al., 2023) | 67.3 | 64.3 | 36.3 | 30.9 | 26.8 |
| StarCoder2-7B (Lozhkov et al., 2024) | 66.9 | 64.1 | 36.6 | 31.1 | 29.4 |
| Gemma-7B (Team et al., 2024) | 65.8 | 62.8 | 34.5 | 29.8 | 27.4 |
| *Closed-Source LLMs* | | | | | |
| Codex (Chen et al., 2021a) | 73.1 | 70.2 | 44.7 | 42.4 | 38.4 |
| ChatGPT (Ouyang et al., 2022) | 71.8 | 68.4 | 44.3 | 40.2 | 38.8 |
| GPT-4 (OpenAI, 2023) | **74.4** | **71.2** | **46.3** | **43.2** | **49.4** |
| *CodeLutra (Ours)* | | | | | |
| Base ($\pi_0$) | 59.3 | 55.1 | 22.3 | 19.5 | 28.2 |
| Iteration 1 ($\pi_1$) | 67.8 | 63.9 | 37.8 | 33.2 | 43.2 |
| Iteration 2 ($\pi_2$) | 72.4 | 68.3 | 40.8 | 36.0 | 46.8 |
| Iteration 3 ($\pi_3$) | **76.6** | **72.5** | **43.1** | **38.6** | **48.6** |
| Iteration 4 ($\pi_4$) | 76.3 | 72.1 | 42.6 | 38.3 | 48.2 |

test cases comprehensively cover edge cases and typical usage patterns to ensure thorough validation. Our correctness verification process is fully automated, allowing efficient processing of thousands of code samples during both training and evaluation phases. For more experimental details, please refer to the Appendix A.1.

## 5.2 Main Results

**Results on the data query task.** We compare CODELUTRA with baselines in code generation for the data query task, as shown in Table 1. We found that existing open-source LLMs like Llama-3-8B still have a significant performance gap in code generation for data queries compared to closed-source LLMs like GPT-4. Although supervised fine-tuning can help bridge this gap—e.g., SFT increases the EX of Llama-3-8B on Spider from 59.3% to 67.9%—there remains a notable difference with GPT-4's 74.4%. Through our refinement framework, Llama-3-8B after four iterations exceeded SFT performance by 16.9% and ***even outperforms GPT-4 with an execution accuracy of 76.6%***. Additionally, on the more challenging BIRD dataset, after three iterations, CODELUTRA significantly improved the EX of the base model from 22.3 to 43.1, achieving performance very close to GPT-4. The similar performance observed between iterations 3 and 4 can be attributed to model convergence, as the model approaches its optimal state with no further gains in later iterations.

**Results on the data science task.** Table 1 also presents results for the data science task, where we evaluate both open-source and closed-source LLMs, as well as our method CODELUTRA. On the DS-1000 dataset, open-source models like Llama-3-8B and Gemma-7B struggle, with significantly lower EM and Pass@1 scores compared to closed-source models like GPT-4. Fine-tuning provides a minor boost in performance, as seen with Llama-3-8B's Pass@1 improving from 28.2% to 30.0%. However, as with the data query task, a large performance gap remains between fine-tuned open-source models and closed-source

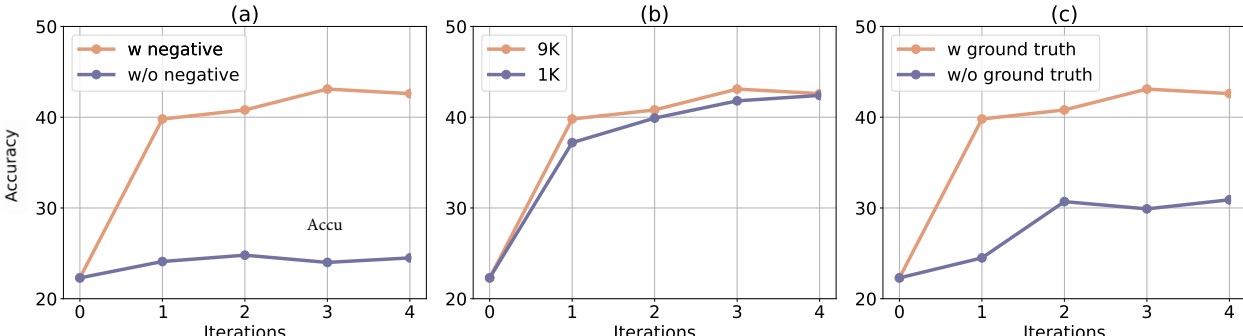

Figure 3: (**a**) Ablations on the effects of negative samples for training. (**b**) Ablations on the question number during training. (**c**) The effects of ground truth for validation during preference datasets collection.

ones, where GPT-4 leads with a Pass@1 score of 49.4%. Nonetheless, CODELUTRA demonstrates substantial improvements (from 28.2% to **48.6%**), offering a promising path for narrowing this gap.

### 5.3 More experiments

**The importance of learning from failed attempts.** Our framework CODELUTRA leverages both positive and negative answer pairs to iteratively improve model performance, particularly by minimizing the generation of incorrect responses. But what happens when we omit the negative samples and rely solely on supervised fine-tuning using positive samples generated by the model? In this ablation, we compare the performance of our objective 4 with a model trained with $\mathcal{L}_{\mathrm{SFT}}(\pi; \mathcal{D}_t^c)$. As seen in Figure 3(a), without negative samples (purple line), the model's performance plateaus across iterations, remaining close to the baseline. In contrast, incorporating negative samples (blue line) leads to steady performance improvements over successive iterations. This ablation confirms that including negative samples is critical to refining the model's ability to distinguish between optimal and suboptimal responses, significantly boosting overall performance.

**CodeLutra achieves strong performance under limited training data.** The cost of acquiring high-quality question-code pairs can be significant, so we examine whether our method truly depends on large datasets. Under the data science code generation task, *we found that using just 500 samples improved Llama-3-8B from an accuracy of 28.2 to 48.6, approaching the performance of GPT-4.* This demonstrates that CODELUTRA achieves strong results even with a limited number of high-quality annotations. We further verify this with the data query task by randomly selecting 1K question-code pairs from BIRD's training data and comparing them to the full 9K sample set. The results, as shown in Figure 3(b), reveal similar trends, with the two setups reaching peak execution accuracies of 43.1 and 42.4, respectively. This minor difference suggests that CODELUTRA does not heavily rely on large volumes of training data and can generalize well with fewer annotations, which is crucial for minimizing the cost of dataset collection.

**Importance of SFT regularization during preference optimization.** Recall in Section 4 that our loss function integrates DPO with SFT to regularize the training, and prevent decreasing likelihood on the correct solution. In Table 2, we ablate the effect of SFT regularization on both the Spider and DS-1000 datasets. Notably, omitting optimizing the SFT loss on the correct solutions results in a marked decline in model performance, *e.g.*, ↓59.4% on Spider. This highlights the ef-

Table 2: Ablations on the SFT on the correct answers.

| Methods | Spider | DS1000 |
|---|---|---|
| $\mathcal{L}_{\mathrm{DPO}}$ | 17.2 | 12.4 |
| Ours | **76.6** | **48.6** |

fectiveness of the dual loss approach, ensuring that the model not only ranks correct solutions higher but also increases their generation probability, leading to more consistent high-quality code outputs.

**CodeLutra remains effective on different base models.** To further validate the generalization capability of our framework CODELUTRA, we extend our experiments to two additional open-source base models: **Gemma-7B** (Team et al., 2024) and **StarCoder-7B** (Lozhkov et al., 2024). As summarized in Table 3, we report the results on both the *Spider* and *DS1000* datasets across multiple iterations of our refinement process. For Gemma-7B, we observe a significant improvement in Execution Accuracy (EX) on Spider, starting from 49.9% at $\pi_0$ (the base model) and reaching 72.6% after four iterations ($\pi_4$). A similar trend is observed in the

| Model | Gemma-7B | | StarCoder-7B | |
|---|---|---|---|---|
| | Spider | DS1000 | Spider | DS1000 |
| $\pi_0$ | 49.9 | 24.2 | 61.2 | 26.8 |
| $\pi_1$ | 63.7 | 38.8 | 72.8 | 39.6 |
| $\pi_2$ | 69.3 | 43.6 | 74.7 | 42.4 |
| $\pi_3$ | 71.3 | 44.4 | 77.2 | 45.2 |
| $\pi_4$ | **72.6** | **44.0** | **77.5** | **45.8** |

Table 3: Performance with CODELUTRA of Gemma-7B and StarCoder-7B across Spider and DS1000 benchmarks.

DS1000 dataset, where the Pass@1 metric improves from 24.2% to 44.0%. For StarCoder-7B, the improvements are also pronounced, with EX on Spider increasing from 61.2% to 77.5%, and Pass@1 on DS1000 rising from 26.8% to 45.8%. These results demonstrate that our framework is robust across different model architectures, consistently yielding significant performance gains regardless of the underlying base model. Notably, the iterative refinement process of CODELUTRA continues to improve the accuracy and correctness of generated code, highlighting the CODELUTRA generalization to different code generation tasks.

# 6 Further Analysis on CodeLutra

**Is ground truth code necessary for preference dataset collection?** Recall that our framework relies on ground truth code to evaluate the quality of generated code during the collection of preference datasets. To test the impact of this dependence, we conduct experiments that replace the ground truth with a more general criterion—whether the generated code is executable. In the absence of ground truth, we consider executable answers as chosen and non-executable ones as rejected. Applying this approach to the Bird dataset, we observe notable gains despite the absence of ground truth: accuracy rose from 22.3 to 30.9 (see Figure 3(c)). Moreover, the proportion of executable code surged from 59.8% to 89.7%, showing that the model effectively learned to avoid common errors, such as syntax issues or missing database tables. *This experiment demonstrates that using executability as a metric still enables substantial model improvements, making the method applicable even without high-quality annotations*, and highlights the robustness of CODELUTRA under such conditions.

**CodeLutra helps reduce the syntax errors across iterations.** To evaluate whether our method enables LLMs to learn from their mistakes over multiple iterations, we sampled 100 error cases from the test set using models $\pi_0, \pi_1, \pi_2, \pi_3$, trained with CODELUTRA on the BIRD dataset for qualitative analysis. We measure the fraction of executable code generated by each model. As shown in Figure 4, the percentage of non-executable code decreases from 40% to 11% when trained with CODELUTRA, indicating that the models have improved in mastering SQL syntax and are better at avoiding basic errors. A qualitative example in Figure 5 highlights this improvement: the base model incorrectly queries the "Currency" column in the wrong table, resulting in an error, while the model trained with CODELUTRA successfully generates the correct SQL query.

**CodeLutra improves quality of incorrect answers across iterations.** Based on the responses of the initial model $\pi_0$, we divide the test set into a correct set and an error set. We track the quality trends of the model in these two sets across iterations. Using the cosine similarity metric based on the BLEURT embedding proposed by (Sellam et al., 2020), we calculate similarities denoted as $\text{sim}\pi_t(\hat{y}, y_{\text{gt}})$ for the model fine-tuned over $t$ iterations. Here $\hat{y}$ denotes the model generation, and $y_{\text{gt}}$ is the ground truth solution. As shown in Figure 4(b), we observe that the similarity between the model's output on the correct set and the ground truth remains stable (see purple bars), while with each iteration, the similarity between the model's output on the error set and the ground truth increases significantly—from 0.48 to 0.54. This indicates that CODELUTRA helps the base model improve outputs on error set, while the outputs on correct cases remain qualitatively stable.

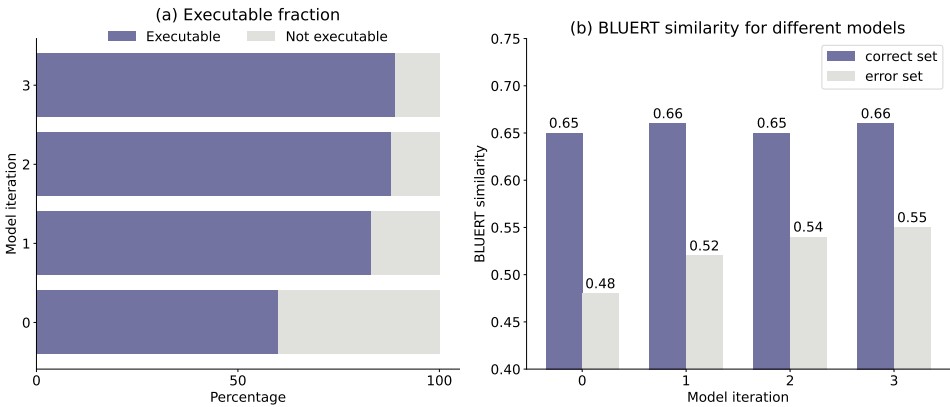

Figure 4: (**a**) Error analysis for different models. (**b**) BLEURT similarity for different models with the ground truth on the correct set and error set divided from the base model.

---

**Different answers from different models for the data query:**

**Database schema:** `CREATE TABLE customers ( CustomerID INTEGER UNIQUE not null primary key, Segment TEXT null, Currency TEXT null ); CREATE TABLE "yearmonth" ( CustomerID INTEGER not null references customers on update cascade on delete cascade, Date TEXT not null, Consumption REAL, primary key (Date, CustomerID) );`
**Question**: Among the customers who paid in euro, how many of them have a monthly consumption of over 1000?
**The answer from $\pi_0$ (wrong):**

```
SELECT COUNT(CustomerID) FROM yearmonth WHERE Consumption > 1000
AND CustomerID IN (
    SELECT CustomerID FROM customers WHERE Currency = 'EUR' );
```

*SQLite error: no such column: Currency*
**The answer from $\pi_3$ (correct):**

```
SELECT COUNT(CustomerID) AS "Number of customers" FROM  "yearmonth"
WHERE  Consumption > 1000 AND    Currency = 'EUR'  GROUP BY CustomerID
```

---

Figure 5: CODELUTRA helps reduce the syntax errors. $\pi_0$ fails due to a missing column, while $\pi_3$ generates the correct query.

# 7 Conclusion

We introduced CODELUTRA, a preference-guided refinement framework that enhances LLMs for code generation using self-generated data from successes and failures, without relying on external datasets or larger models. Experiments show CODELUTRA significantly improves base LLMs, with Llama-3-8B outperforming GPT-4 on data query tasks and nearly matching its performance on data science tasks with limited training data. Additionally, CODELUTRA reduces coding errors while improving code quality and accuracy, offering a cost-efficient, scalable solution for LLM code generation.

## Ethics Statement

Our framework CODELUTRA enhances code generation by leveraging both successful and failed attempts. However, it may propagate biases from pretrained datasets, leading to unfair or discriminatory outputs. There is also a risk of misuse for generating harmful code, and we advocate for its responsible application. Additionally, the approach relies on self-generated data, which may inadvertently amplify errors if not carefully validated. We encourage users to apply CODELUTRA ethically, ensuring fairness, safety, and high-quality outcomes.

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

# A   More information

## A.1   Experimental setup

Table 4 summarizes the training hyperparameters used for data query and data science tasks across each iteration. It includes key training parameters such as learning rate, batch size, LoRA rank, etc. We set

Table 4: Summary of training hyperparameters for data query and data science for each iteration.

|  | Parameters | Value |
|---|---|---|
| Data query | Number of epochs | 1 |
| | Learning rate | $5 \times 10^{-5}$ |
| | $\beta$ | 0.1 |
| | Batch size | 16 |
| | Gradient accumulation steps | 1 |
| | Maximum sequence length | 2048 |
| | DeepSpeed Zero stage | 2 |
| | Weight decay | 0.0001 |
| | LoRA rank | 8 |
| | $\lambda$ | 1.0 |
| Data science | Number of epochs | 1 |
| | Learning rate | $5 \times 10^{-5}$ |
| | $\beta$ | 0.5 |
| | Batch size | 16 |
| | Gradient accumulation steps | 1 |
| | Maximum sequence length | 512 |
| | DeepSpeed Zero stage | 2 |
| | Weight decay | 0.0001 |
| | LoRA rank | 8 |
| | $\lambda$ | 0.5 |

$K$=10 for each iteration, generating 10 positive and negative sample pairs per question. So for the training set of the preference learning in the data science task, we collect about **5k** samples and **100k** for the data query task. To maintain quality when selecting incorrect samples, we filter out answers that contain repeated strings.

## A.2   Datasets

*Spider* includes 10,181 questions with 5,693 unique SQL queries across 200 databases in 138 domains, while *BIRD* contains 12,751 question-SQL pairs across 95 large databases, covering over 37 domains. We utilize *DS-1000*, which comprises 1,000 data science problems sourced from Stack Overflow, covering seven Python libraries related to analysis in data science. The dataset is designed to minimize memorization risk by modifying original problems and uses a multi-criteria evaluation system to assess functional correctness and coding constraints. We split DS-1000 into 500 samples for training and 500 for evaluation.

## A.3   Examples for different datasets

For the Data Query task, the model is given a natural language problem description and is tasked with generating the corresponding SQL query for a database using an LLM. For example, given the description "*How many heads of the departments are older than 56?*", the model should produce the appropriate SQL query to execute this request. In the Data Science task, the LLM is tasked with generating the correct Python code to solve a given data science problem. For instance, given the problem "*I have a 2D array to represent a many-many mapping. What is the quickest way to zero out the second row and the first column?*", we test

LLM's ability to solve data science problems with `numpy`. We provide examples that highlight the tasks used to evaluate our framework. The first example illustrates the Data Query task (see Figure 6), where models generate SQL queries from natural language descriptions based on a given database schema. The second example showcases the Data science task (see Figure 7), in which models write Python code to solve typical data manipulation problems, such as processing a data frame. These examples reflect common real-world applications of language models in both querying databases and performing data science operations.

### A.4 Qualitative example

Figure 5 illustrates how CODELUTRA refines incorrect attempts into correct solutions over successive iterations. In the shown scenario, the model is tasked with querying a database for the number of customers paying in euros who have monthly consumption over 1000. Initially, the baseline model $\pi_0$ generates an SQL query that references a non-existent column `Currency` directly in the `yearmonth` table, causing an execution error. By the third iteration, the refined model $\pi_3$ has learned to correctly integrate the `Currency` condition while preserving syntactic and semantic integrity. It accurately groups results by `CustomerID`, retrieves the correct subset of customers, and computes the intended count. This example highlights CODELUTRA's ability to iteratively overcome syntactic and logical pitfalls without explicit feedback, resulting in more reliable and accurate code generation over time.

### A.5 Model checkpoints and reproducibility

To ensure reproducibility of our experiments, Table 5 provides the exact model checkpoints used for each model in our evaluation. For proprietary models, we include the specific version and access method.

Table 5: Exact model checkpoints used in our experiments.

| Model | Checkpoint/Version |
|---|---|
| Llama-3-8B | meta-llama/Llama-3-8B-Instruct |
| Llama-3-70B | meta-llama/Llama-3-70B-Instruct |
| CodeLlama-7B | codellama/CodeLlama-7b-Instruct-hf |
| StarCoder-15B | bigcode/starcoder |
| Gemma-7B | google/gemma-7b-it |
| Codex | code-davinci-002 (via Azure OpenAI Service) |
| GPT-3.5 | gpt-3.5-turbo-0613 |
| GPT-4 | gpt-4-0613 |

### A.6 Setup of the baseline

To ensure an apple-to-apple comparison, all experiments report performance metrics using the same prompt, guaranteeing the comparability of the model evaluation results. For the sampling hyperparameters during evaluation, we set the temperature to 0.2, the maximum tokens to 512, and top-p to 1.0. The SFT model reported in Table 1 consists of training for 1 epoch with a learning rate of $5 \times 10^{-5}$, a batch size of 16, and gradient accumulation steps set to 1. Regularization is applied with a weight decay of 0.0001, while a LoRA rank of 8 is employed to enable low-rank adaptation. Additionally, the setup includes $\beta = 0.5$ and $\lambda = 0.5$ to balance specific model training.

## B More results

### B.1 Assessment ability

Models trained with the DPO loss are capable of assessing the quality of code answers. To prevent data leakage, we utilized the robust open-source model Codestral to generate multiple samples on Bird's test set, constructing positive and negative sample pairs based on execution accuracy. We evaluated the fine-tuned

---

**Different answers from different models for the data query:**

**Database schema:**

```
CREATE TABLE customers ( CustomerID INTEGER  UNIQUE not null

primary key, Segment TEXT null, Currency TEXT null );

CREATE TABLE gasstations ( GasStationID INTEGER UNIQUE not null primary key,

ChainID INTEGER null, Country TEXT null, Segment TEXT null );

(Omit other database information...)
```

**Ground truth another:**

```
SELECT T2.Consumption  FROM transactions_1k AS T1
INNER JOIN yearmonth AS T2 ON T1.CustomerID = T2.CustomerID
WHERE T1.Price / T1.Amount > 29.00
AND T1.ProductID = 5   AND T2.Date = '201208';
```

---

Figure 6: An example from the data query dataset from the BIRD (Li et al., 2024).

---

**An example of data science from the DS1000 (Lai et al., 2023):**

**Problem:**

```
I have a simple dataframe which I would like to bin for every 4 rows.

It looks like this:

col1\n0    1\n1    1\n2    4\n3    5\n4    1\n5    4\n

and I would like to turn it into this:

col1\n0    11\n1    5\n

I have already posted a similar question here

but I have no idea how to port the solution to my current use case.

Can you help me out?
```

**Solution:**

```python
def g(df):
    return df.groupby(df.index // 4).sum()

result = g(df.copy())
```

---

Figure 7: An example of data science from the DS1000 (Lai et al., 2023).

---

LLM's ability to accurately assess code quality by measuring the classification accuracy on this dataset. Under the standard supervised fine-tuning (SFT) setting, the model achieved a classification accuracy of 56%, which is close to random guessing and indicates that SFT alone lacks this capability. In contrast, our CODELUTRA attain a classification accuracy of 79%, demonstrating that our approach enables the model to better understand code characteristics and select correct answers. This substantial improvement highlights the potential of CODELUTRA.

Table 6: Code quality assessment accuracy.

| Methods | Accuracy (%) |
| --- | --- |
| Supervised fine-tuning | 56.3 |
| Preference learning | 79.6 |

## B.2 Comparison of execution-based methods

To examine the limitations of previous methods on small LLMs, we evaluated SFT, SFT with self-debugging (Chen et al., 2024), and CodeLutra on the Spider benchmark using Llama-3-8B as the base model. The results are summarized in Table 7. While self-debugging offers a minor improvement over SFT (+1.3%),

| Method | Execution Accuracy (%) |
| --- | --- |
| SFT | 67.9 |
| SFT + Self-debug | 69.2 |
| CodeLutra (Ours) | 76.3 |

Table 7: Execution accuracy on the Spider benchmark with Llama-3-8B.

CodeLutra outperforms both, achieving 76.3% execution accuracy. This highlights its ability to leverage execution feedback effectively through probabilistic comparisons, iteratively refining model performance without relying on complex reward signals or handcrafted prompts. These results demonstrate the scalability and generalizability of CodeLutra, even with smaller, open-source models.

## B.3 Comparision with GPT-4 preference on code

To assess the reliability of using execution results versus GPT-4 preferences for constructing pairwise preferences, we conducted experiments on DS-1000 with Llama-3-8B as the base model. Specifically, we used GPT-4 to judge which answer in each pair is better, similar to the approach employed by RLAIF (Lee et al., 2023). The results are presented in Table 8.

| Preference Source | Accuracy (%) |
| --- | --- |
| GPT-4 Preference | 27.8 |
| CodeLutra (Execution Results) | 48.2 |

Table 8: Comparison of accuracy using GPT-4 preferences versus execution results for constructing pairwise preferences on DS-1000.

The results indicate that using execution results, as implemented in CodeLutra, achieves a significantly higher accuracy (48.2%) compared to GPT-4 preferences (27.8%). This demonstrates that execution-based signals provide more reliable guidance for refining models, particularly for smaller LLMs like Llama-3-8B. In contrast, GPT-4 preferences appear to struggle, potentially due to the model's lack of grounding in execution semantics and over-reliance on heuristic judgments.

## B.4 Additional Benchmark Results

To demonstrate the generalizability of our approach beyond the primary datasets used in our main experiments, we evaluated CODELUTRA on the MBPP benchmark from EvalPlus (Liu et al., 2023a). Using Llama-3-8B as our base model, we compared the performance of the base model, standard supervised fine-tuning (SFT), and our CODELUTRA approach.

These results confirm that CODELUTRA's effectiveness extends beyond our primary datasets, demonstrating its broad applicability across different code generation tasks and benchmarks.

| Methods | Accuracy (%) |
|---|---|
| Base | 52.3 |
| SFT | 53.1 |
| CodeLutra | 55.4 |

Table 9: Performance comparison on the MBPP benchmark from EvalPlus using Llama-3-8B.

## C   Limitation and future work

While CODELUTRA enhances code generation performance by leveraging self-generated data, it exhibits limitations that warrant consideration. The framework focuses on the correctness of the generated code, overlooking other vital aspects such as efficiency, readability, and adherence to specific formal specifications, which are essential for practical applications. Additionally, CODELUTRA treats all failed code attempts uniformly, without distinguishing between different types or severities of errors, potentially limiting the model's ability to learn from more informative mistakes. Furthermore, our current sampling strategy for preference pairs could be refined—while we sample with replacement when there are fewer than K examples, statistical theory suggests that sampling with replacement even when more examples are available may be beneficial, as it would better approximate sampling from the underlying distribution (similar to bootstrapping in statistics). To address the aforementioned limitations, future research related to CODELUTRA should explore several key directions. Expanding the preference-guided refinement mechanism to incorporate additional criteria such as code efficiency and compliance with formal specifications would enhance the overall quality and utility of the generated code. Developing a more nuanced approach to categorizing and prioritizing failed code attempts based on the type and severity of errors could enable more targeted and effective learning, thereby improving the model's ability to avoid similar mistakes in future generations. Exploring alternative evaluation methods, such as static code analysis or formal verification tools, could reduce the framework's reliance on execution results and broaden its applicability to a wider range of tasks. Additionally, investigating optimal sampling strategies for preference pairs, including consistent sampling with replacement regardless of sample size, may further improve the statistical properties of our approach.

