# OpenReview forum: "CodeLutra: Boosting LLM Code Generation via Preference-Guided Refinement"
_TMLR — Accepted by TMLR_

### Review · Reviewer_Jkh2 · 2025-03-25

**Summary Of Contributions:**

This paper investigates the use of preference-based reinforcement learning for code generation. The core idea is to sample multiple generations from the model during training, classify them into positive and negative examples based on whether they pass the unit tests, and to then use DPO to optimize the likelihood of generating positive samples. The authors use this approach to finetune models based on Llama-3-8b, Gemma-7b and StarCoder-7b on two SQL and one Python data science datasets. They find that with this approach, the models approach the performance of larger closed-source models. Furthermore, through a series of ablations the authors show that the use of preference learning on positive vs. negative examples itself is a critical part of the pipeline, and that doing simple expert iteration (SFT on self-generated positive examples) does not work nearly as well.

**Audience:**

Yes

**Broader Impact Concerns:**

None.

**Claims And Evidence:**

Yes

**Requested Changes:**

- (Critical): Address the possible confounding factor mentioned above. I think it is important to investigate this further, even if just by doing some manual inspection of the data you have already generated (e.g. you may not need to run any new experiments).
- (Critical): Justify your choice of datasets over other established choices such as MBPP, HumanEval(-++), and LiveCodeBench.
- (Critical): Please fix your citations. There are multiple inferences of using \cite instead of \citep, leading to statements like "While previous approaches Le et al. (2024)" instead of "While previous approaches (Le et al., 2024)".
- (Critical): Please add (in the appendix) a table of the exact checkpoints you use for each model. This is important for reproducibility. I am also a bit surprised to see Codex in your experiments given it's now been about 2 years since it was officially retired from the OpenAI API, so to aid reproducibility, please include a statement on how you obtained access to the model (is it still served on Azure?).
- (Critical): Related to the above, please clarify what you mean by "ChatGPT" in the text. "ChatGPT" is not a model; it is a web interface. I presume you mean some checkpoint of GPT-3.5, but is not clear which since the models being served on chatgpt have evolved substantially since its initial launch.
- (Critical): In the last paragraph on page 10 there is an incorrect figure reference: Figure 3(b) should be Figure 4(b).
- (Not critical): Tone down the language in the abstract and introduction. I think the paper is very solid and interesting, but to me the current language is a bit over the top.
- (Not critical): Intuitively I expect you should probably be sampling with replacement even when you have more than K examples. This is because, in expectation over the draw of samples, sampling with replacement is equivalent to sampling from the base distribution (see the concept of bootstrapping in statistics). This is a minor point, and doesn't affect the overall validity of your empirical claims, but I thought it was worth mentioning in case you want to consider it in future work.

**Strengths And Weaknesses:**

Strengths:

- The paper is well written and very easy to follow. Figure 1 gives an exceptionally good overview (although it looks a bit garish on my monitor), and the methodology and experiments are presented very clearly.
- The authors have carried out a fairly thorough set of experiments. I especially appreciate the ablations in Figure 3 as well as the fact that the results were replicated for 3 base models.
- The methodology is very straight forward; essentially this is just applying DPO to code generation, and it is interesting to see that this works well given that historically speaking RL for code generation has not been able to live up the expectations.

Weaknesses:

- The choice of datasets is a bit weird to me. SQL and Python code generation is a fine setting, since you can expect to have good test coverage, but I expected to find more established datasets like HumanEval(-++), MBPP, or LiveCodeBench. I think the choice of datasets requires some justification by the authors.
- The abstract and introduction oversell the novelty of the paper. I don't really see the point of giving your method a new name when it is essentially just taking an old idea (RL for code gen), using a newly developed method (DPO), and finding that it does surprisingly well. To clarify I think that is a very solid contribution to the literature, and definitely of interest to the community, and I understand the desire to give your framework a simple and rememberable name. I just think some of the language in the abstract and introduction is a bit over the top: instead of saying "We present CodeLutra, a novel framework that iteratively improves LLMs for code generation [...]" you could have just said "We apply recent developments in preference-based post-training to code generation, and find that it works surprisingly well".
- One hypothesis for the trends you observe is that there are flaws in the execution framework for the benchmarks that are artificially limiting which generations are deemed "correct".
For example, it may be that the execution engine does not allow for imports of external libraries, or that it does not allow for the use of certain language features, or simply that it expects the answers to be given in a certain format (e.g. defining and calling a function with a particular name vs. top-level scripting). This would be a major confounder in your results, since the large performance gains you observe may then simply boil down to the model learning to avoid these features; indeed this might explain your findings in Figure 3(c), and potentially the very large gap between execution feedback and RLAIF in Table 7. I do not think the paper in its current form has taken enough steps to investigate the impact of this confounding factor. I would recommend that consider doing some more extensive qualitative/manual inspection of the generations, or consider extending your core findings to a broader set of domains. As mentioned above, in particular I would encourage the authors to consider LiveCodeBench, which is a more recent code generation benchmark.

---

> ### Author Response · Authors · 2025-04-16
> **Response to Reviewer Jkh2**
>
> > W1 & Q2: Justification of the dataset selection and more results to verify the generalization.
>
> We selected DS-1000 and SQL tasks because they represent common data analysis and data query scenarios that developers encounter daily. These datasets: (1) provide robust test cases for execution verification, (2) focus on practical usage in the real world for the AI assistant rather than purely algorithmic problems, and (3) cover diverse programming patterns across multiple domains. This complements benchmarks like HumanEval that primarily test algorithmic reasoning.
>
> To demonstrate generalizability, we tested CodeLutra on MBPP from EvalPlus [1] using Llama-3-8B:
>
> | **Methods**          | **Accuracy** |
> |-----------------------|-------------------------|
> | Base                  | 52.3%                  |
> | SFT                   | 53.1%                  |
> | CodeLutra             | 55.4%                  |
>
> These results confirm that CodeLutra's effectiveness extends beyond our primary datasets.
>
> [1] Liu, Jiawei, et al. "Is your code generated by chatgpt really correct? Rigorous evaluation of large language models for code generation." Advances in Neural Information Processing Systems 36 (2024).
>
> > W2 & Q6: The reframing of the abstract.
>
> Thank you for recognizing our contribution. We have revised the abstract and introduction according to your comments.
>
> > W3 & Q1: Execution framework limitations as a confounder
>
> Thank you for raising this important concern. We've investigated this potential confounder through manual inspection of model generations. Our analysis reveals that execution failures primarily stem from genuine logical errors and syntax issues rather than framework limitations. For SQL tasks specifically, we observed common errors related to schema linking and misunderstanding database content, which are substantive problems in code generation rather than artifacts of the evaluation framework. This error analysis is also done in previous work [1].
>
> The benchmarks we used are designed to allow necessary imports and provide clear specifications. We've included additional evaluation results on MBPP in Appendix B.4, which show consistent improvements and further validate that our approach addresses genuine code generation challenges across different domains.
>
> [1] Li, Jinyang, et al. "Can LLM already serve as a database interface? A big bench for large-scale database grounded text-to-sqls." Advances in Neural Information Processing Systems 36 (2023): 42330-42357.
> >Q3: format of the citation.
>
> Thanks for pointing out the issue. We already checked it carefully and corrected these errors.
>
> > Q4: Checkpoints usage.
>
> We appreciate this important point about reproducibility. We already added the "Model checkpoints and reproducibility" in Section Appendix A.5 We have added a detailed table in the appendix that specifies the exact checkpoints used for each model in our experiments. Regarding Codex, we accessed it through Azure OpenAI Service, which still provides the model even after it was retired from OpenAI's main API. We will clarify this access method in the paper to ensure reproducibility. Regarding ChatGPT, we should clarify that we specifically used the GPT-3.5-Turbo model (gpt-3.5-turbo-0613) for our experiments. We acknowledge that referring to it as "ChatGPT" is imprecise, and we will update all references to specify the exact model version throughout the paper.
>
> > Q5: In the last paragraph on page 10 there is an incorrect figure reference: Figure 3(b) should be Figure 4(b).
>
> Great catch - we corrected this error.
>
> > Q7: Future work related to bootstrapping.
>
> Thanks for the suggestion! We think this is a good suggestion and we have added it to future work.

---

> > ### Comment · Reviewer_Jkh2 · 2025-04-22
> >
> > Thank you for your answers and for updating the manuscript. I have no further questions at this time.

---

### Review · Reviewer_EmsM · 2025-04-07

**Summary Of Contributions:**

This paper uses iterative preference-based refinement to improve code llms, which achieve good performance.

**Audience:**

Yes

**Claims And Evidence:**

Yes

**Requested Changes:**

In Table 1, the bolded number needs to be corrected.

I suggest elaborating on how the code generation problem differs from related tasks in previous work to better highlight the novelty, as the main methodology is not novel enough.

It would be helpful to further illustrate/introduce the phenomenon where the model tends to decrease the likelihood of both correct and rejected code during training in detail.

I recommend providing more details on how the test inputs are generated, including how reference code is accessed and how the inputs are designed to meaningfully compare the functionality of the generated code with the reference.

**Strengths And Weaknesses:**

Strengths:
The experimental results are impressive.


Weaknesses:
The novelty is limited, as the combination of DPO and SFT loss functions has been previously explored, and the iterative refinement approach is also not new.

---

> ### Author Response · Authors · 2025-04-16
> **Response to Reviewer EmsM (1/2)**
>
> We thank the reviewer for valuable suggestions for improving the quality of our work. Before addressing the concerns, we would first kindly refer R2 to the review guideline for TMLR:
>
> > ...The novelty of the studied method is not a necessary criterion for acceptance. **We explicitly avoid these terms (“significant”, “impactful”, “novel”), and focus instead on the notion of “interest”**.
>
> However, with that being said, we are still happy to clarify the significance and novelty of our work. While we agree with the reviewer that iterative refinement has been explored in other contexts such as mathematical reasoning, **our work is the first to explore such technique for code generation—a domain with fundamentally different challenges**. Below we highlight our distinct contributions:
>
> 1. **Code generation presents unique requirements and challenges** that differ substantially from mathematical reasoning, such as execution correctness, syntax precision, and logical consistency. For code generation, even minor errors can render the entire solution non-functional, making the task significantly more sensitive than reasoning tasks. **It's important to note that methods effective for mathematical reasoning do not necessarily transfer to code generation**. For example, our experiments show that simply using DPO yields poor results for code (see Table 2), while it performs well for reasoning tasks. Our paper demonstrates phenomena specific to code generation that provide valuable guidance for future research in this area.
>
> 2. **CodeLutra focuses on leveraging the correctness of generated code rather than the feedback information itself**. This distinguishes our work from previous methods that often directly incorporate feedback (e.g., error messages or debugging hints) to guide improvements. Our approach avoids overfitting to specific error patterns and ensures broader generalizability by iteratively refining outputs using both successful and failed generations—a novel contribution to the field.
>
> 3. **In-depth understanding**: Our method goes beyond simply applying iterative refinement to a new scenario. Section 4 and our analyses highlight CodeLutra's insights unique to code generation that are not presented in [1]:
>    - **Failed Attempts Matter**: Incorporating failed attempts in training leads to significant performance improvements in code generation, achieving results comparable to or exceeding leading models.
>    - **Applicability without ground truth**: Even without high-quality annotations, using executability as a metric enables effective refinement, reducing common errors like syntax issues, which remarkably improves code generation performance. **This aspect is not explored in reasoning tasks**.
>    - **Improving incorrect answers**: Our analysis shows that CodeLutra progressively enhances the quality of incorrect answers across iterations. Specifically, the similarity between the model's outputs on the error set and the ground truth increases significantly, as illustrated in Figure 3(b). Simultaneously, the similarity for correct answers remains stable, ensuring that improvements focus on error cases without degrading already accurate outputs.
>
> 4. **Results demonstrating real-world impact**: Our framework achieves state-of-the-art results, matching or surpassing GPT-4 on specific metrics, using only limited training data. This demonstrates the practical and scientific significance of CodeLutra in democratizing access to high-performing LLMs for code generation tasks.
>
> These distinctions emphasize that CodeLutra addresses the specific challenges of code generation with a principled and scalable approach, making it fundamentally different and more specialized than previous work.
>
> [1] Pang, Richard Yuanzhe, et al. "Iterative reasoning preference optimization." arXiv preprint arXiv:2404.19733 (2024).
>
> > Requested change 1: In Table 1, the bolded number needs to be corrected.
>
> Thanks for catching this issue! We have fixed this in Table 1.

---

> > ### Author Response · Authors · 2025-04-16
> > **Response to Reviewer EmsM (2/2)**
> >
> > > Requested change 2: Elaborating on how the code generation problem differs from related tasks in previous work to better highlight the novelty, as the main methodology is not novel enough.
> >
> > We appreciate the comment. **As suggested, we have expanded the novelty discussion in Section 2 and Section 3, emphasizing differences between code generation and other tasks**. Code generation differs from many NLP tasks in that it requires strict syntactical and semantic precision and its correctness can be directly verified via execution. This deterministic feedback enables our framework to effectively identify and correct specific error modes—such as syntax errors or logical flaws—that are less apparent in other tasks.  A key challenge unique to code generation is the "locally correct but globally incorrect" phenomenon, where code with high token-level probability can still be functionally incorrect (e.g., a single misplaced character can break otherwise perfect code). Traditional likelihood-based methods struggle with this binary nature of code correctness. CodeLutra addresses this challenge through its execution-guided iterative refinement process that leverages both successful and failed generations, representing a novel application of preference optimization specifically tailored to code generation's unique requirements.
> >
> > > Requested change 3: It would be helpful to further illustrate/introduce the phenomenon where the model tends to decrease the likelihood of both correct and rejected code during training in detail.
> >
> > As suggested, we have explained further the phenomenon of likelihood drop for both correct/rejected codes in DPO using insights from Liu et al. (2024). Please see the updated manuscript for changes.
> >
> > [1] Liu, Zhihan, et al. "Provably mitigating overoptimization in rlhf: Your soft loss is implicitly an adversarial regularizer." arXiv preprint arXiv:2405.16436 (2024).
> >
> > > Requested change 4: Providing more details on how the test inputs are generated.
> >
> > Thank you for this suggestion. For test input generation and functionality comparison, we employ different approaches based on the task type. For data query tasks, we execute SQL queries against the same database schema and compare execution results directly. This ensures that the generated queries retrieve the same data as the reference solutions when run against identical database states. For data science tasks, we leverage the DS-1000 evaluation framework, which provides standardized test cases specifically designed to verify functional equivalence between generated and reference code. These test cases comprehensively cover edge cases and typical usage patterns, ensuring thorough validation of code functionality. The framework automatically executes both the generated code and reference solutions with identical inputs and compares their outputs. Our correctness verification process is fully automated, allowing efficient processing of thousands of code samples during both training and evaluation phases. The verification focuses on functional equivalence rather than syntactic similarity, meaning that differently structured but functionally identical solutions are correctly identified as equivalent. **We've included more detailed information about our experimental setup in Section 5.1**.

---

### Review · Reviewer_LfPu · 2025-04-07

**Summary Of Contributions:**

This work introduces a new way of finetuning LLMs by combining the Direct Preference Optimization (DPO) loss and the supervised finetuning (SFT) loss. The method is applied to code execution benchmarks, notably Data Query and Data Science benchmarks.
The method consists of sampling M solutions per problem with the current model, making a dataset of input & (correct, wrong) outputs; and updating the model weights based on the DPO - $\lambda$ SFT loss.

Experiments show that this method produces models with performance comparable to large, closed source models such as GPT4 and outperform finetuned opensource models.

Ablation studies confirm the importance of the SFT term in the proposed objective, the importance of negative examples, and the effectiveness of the approach on different base models (llama, StarCoder, Gemma).

Further analysis shows that the proposed method helps reduce syntax errors and improves the quality of incorrect answers across iterations.

**Audience:**

Yes

**Broader Impact Concerns:**

Broad Impacts discussed in the paper are good.

Maybe could add a sentence about the impact that better code models can have on the job market, especially for software engineers.

**Claims And Evidence:**

Yes

**Requested Changes:**

Nothing major.

Small things:
- Explain why Model@4 < Model@3 in Table 1
- Put a Y axis label on Figure 3

**Strengths And Weaknesses:**

# Strengths

This work proposed a novel way to combine DPO and SFT objectives for Code Generation models. The paper is well written and flows naturally. Experiments are thorough, show strong performance, and ablation studies highlight the importance of positive & negative examples as well as the SFT regularization. The method is also tested successfully on various base models.

# Weaknesses

The approach seems generic enough to be applied to any rule-based environment, not only code. The contribution feels more like a DPO regularization than a code generation-specific method. As such, it would be nice to experiment on other domains with rule-based rewards, and potentially remove the name “Code” from “CodeLutra”.

Can the authors comment on why, in Table 1,  the model at iteration 4 is performing slightly lower than the model at iteration 3? This should be discussed in the main text also.

---

> ### Author Response · Authors · 2025-04-16
> **Response to Reviewer LfPu**
>
> We sincerely appreciate your valuable feedback and insightful questions. Below, we provide detailed responses to each of your points.
>
> > W1: The approach may apply to other rule-based environments beyond code; suggestion to experiment in other domains and reconsider the name.
>
> We appreciate the comment and agree that the underlying mechanism is generic. However, our focus is on code generation—where execution feedback (i.e., whether code runs correctly) plays a crucial role—and our dual-loss design is tailored to handle the strict syntactic and semantic requirements of code. Thus, while the method could extend to other rule-based environments, we maintain “Code” in CodeLutra to emphasize our focus on improving LLM code generation. We also see this broader applicability as a promising direction for future work.
>
> > W2: Explanation of the performance decrease between the epochs 3 and 4.
>
> The slight performance degradation observed between iterations 3 and 4 can be attributed to diminishing returns and stochastic variations during model convergence. After iteration 3, improvements naturally plateau as the model approaches its optimal state. At this stage, minor fluctuations due to training noise can cause temporary performance dips—a common phenomenon in iterative refinement processes. This pattern aligns with established observations in machine learning where performance gains become increasingly marginal as training progresses.
>
> > Minor: adding a Y-axis label to Figure 3
>
> We have fixed this - thank you for the suggestion!

---

> > ### Comment · Reviewer_LfPu · 2025-04-18
> > **acknowledgement of revision**
> >
> > Thank you for the updated manuscript. I have read and acknowledged the updates. No further questions at this time.

---

### Decision · Action_Editor_fEr3 · 2025-05-04

**Recommendation:** Accept with minor revision

**Comment:**

This paper proposes a finetuning methodology for code generation based on preferences of correct against incorrect code outputs. It achieves this through Direct Preference Optimization (DPO) by comparing successful and unsuccessful outputs to align LLMs to correct outputs, while also combining it with supervised finetuning (SFT) to handle the syntactic and semantic requirements of code.

Strengths
- Experiments are solid and comprehensive to demonstrate the improvements for the proposed method over a variety of backbone models (Reviewer LfPu, Reviewer EmsM, Reviewer Jkh2),.
- Ablation study illustrates the roles of individual pieces of design including positive vs negative examples and SFT (Reviewer LfPu, Reviewer Jkh2).
- The paper is generally well written and the presentation is easy to follow (Reviewer LfPu, Reviewer Jkh2).

Weaknesses
- The reviewers have mixed opinions over the novelty of the proposed method (Reviewer LfPu deems this novel, while Reviewer EmsM and Reviewer Jkh2 point out the novelty as a limitation of this paper.)
- The approach design seems generic and thus it is not clear how it is different from other related tasks (Reviewer LfPu, Reviewer EmsM).
- More details can be provided such as training dynamics and test input generation (Reviewer EmsM ), and the execution framework limitations as a confounder (Reviewer Jkh2).

Overall, this is a solid empirical contribution to the code generation for open-source models. Author responses generally resolve the questions from the reviewers. Please incorporate the discussion and required changes by reviewers in the final version, especially the additional results of MBPP.

**Audience:**

Yes

**Claims And Evidence:**

Yes